# Ocean Plastic Assimilator v0.2 : Assimilation of Plastics Concentration Data Into Lagrangian Dispersion Models.

Axel Peytavin[1], Bruno Sainte-Rose[1], Gael Forget[2], and Jean-Michel Campin[2]

[1]The Ocean Cleanup, Batavierenstraat 15 4-7th floor 3014 JH Rotterdam The Netherlands
[2]Massachusetts Institute of Technology, Dept. of Earth, Atmospheric and Planetary Sciences

**Correspondence:** Axel Peytavin (a.peytavin@theoceancleanup.com)

**Abstract.** A numerical scheme to perform data assimilation of concentration measurements in Lagrangian models is presented, along with its first implementation called Ocean Plastic Assimilator, which aims at improving predictions of plastics distributions over the oceans. This scheme uses an ensemble method over a set of particle dispersion simulations. At each step, concentration observations are assimilated across the ensemble members by switching back and forth between Eulerian and

Lagrangian representations. We design two experiments to assess the scheme efficacy and efficiency when assimilating simulated data in a simple double gyre model. Analysis convergence is observed with higher accuracy when lowering observation variance or using a circulation model closer to the real circulation. Results show that the distribution of plastics mass in an area can effectively be improved with this simple assimilation scheme. Direct application to a real ocean dispersion model of the Great Pacific Garbage Patch is presented with simulated observations, which gives similarly encouraging results. Thus, this

method is considered a suitable candidate for creating a tool to assimilate plastics concentration observations in real-world applications to estimate and forecast plastics distributions in the oceans. Finally, several improvements that could further enhance the method efficiency are identified.

## 1  Introduction

Plastic pollution reveals itself to be an urgent matter if humans are to preserve their oceans. Previous publications such as

Lebreton et al. (2018) reviewed how plastics are rapidly accumulating in the oceans and concentrate in oceanic gyres. As public and private ventures set out cleanup goals, accurate and regular forecasts of the state of plastics in the oceans become necessary.

A modeling framework is currently undergoing development at The Ocean Cleanup towards this goal, as the company set itself out to clean 90% of the oceans floating macroplastics by 2040. It is used to assess and improve our ability to perform the

largest cleanup in history.

This framework, of which results are presented in Lebreton et al. (2018), is built upon the Pol3DD Lagrangian dispersion model and presented in Lebreton et al. (2012). In this model, virtual particles representing plastics are generated and let drift over time using currents data extracted from the oceanic circulation modeling system HYCOM (HYbrid Coordinate Ocean

Model, see Bleck (2002)). Results from this model are compared with two other plastic forecast models in van Sebille et al. (2015).

While the Lebreton et al. (2012) modeling framework has already produced valuable results, it is not able to assimilate observations and update forecasts accordingly yet. However, as the company prepares to release a number of systems to clean the Ocean, it will soon dispose of numerous sources of data collecting devices measuring plastics concentration in the oceans. Therefore, we believe it is timely to develop a method to assimilate incoming real-time observations.

Methods to assimilate plastics concentration observations over a Lagrangian dispersion model are in the early development stage (Lermusiaux et al. (2019)). However, earlier studies dealing with data assimilation applied to the atmospheric dispersion of particles around polluting facilities, such as Zheng et al. (2007), have been published.

This paper introduces Ocean Plastic Assimilator v0.2, a numerical scheme developed to assimilate plastics concentration data into 2D Lagrangian dispersion models. Section 2 formulates the method and section 3 then describes its initial implementation and application. For this proof-of-concept paper, we use a dispersion simulation generated with the OpenDrift framework in a controlled environment based on a double gyre analytical flow field. The assimilation results are presented in section 4. Real-world application perspectives and future developments that could further improve the method are discussed in section 5. Finally, in section 6, we present a direct application of the method to a dispersion model of the actual Great Pacific Garbage Patch, with simulated observations sampled from another simulation.

## 2    Method

This section formulates our methodology to perform data assimilation of plastics concentration (or density) observations in any 2D Lagrangian dispersion model, using an Ensemble Kalman Filter (EnKF). It includes: the two representations of data (Eulerian and Lagrangian) being used for this process, the transformation between Eulerian and Lagrangian space, the ensemble assimilation method itself, and model ensemble initialization.

### 2.1    Representations of data

The distribution of plastics mass in a Lagrangian dispersion model is represented through weighted particles drifting according to a flow field in a 2D domain. Each virtual particle represents a drifting plastics concentration. In turn, virtual concentration measurements are collected at fixed locations (grid points) within the studied 2D domain, i.e. an Eulerian representation of the plastics mass distribution.

Our method aims to assimilate concentration observations collected in the Eulerian representation and update the Lagrangian representation accordingly. One cycle of this process consists of projecting particle weights on the concentration grid, assimilating observation data into the concentration grid, projecting grid cell concentration updates on particle weights, and finally letting particles drift until the next assimilation time step. This procedure is summarized in figure 1.

The complete workflow requires:

– An assimilation method.

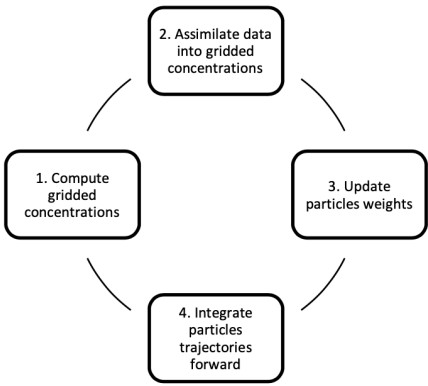

**Figure 1.** Schematic depiction of the 4 steps of our method

- – A dispersion model along with the flow field used.

- – Projections methods to go back and forth between Eulerian and Lagrangian representations.

- – Prior estimates for model parameters and uncertainties.

### 2.2 Procedure

This section presents our procedure on a set of $N_p$ particles drifting in a gridded domain, with a grid size $(m,n)$, and indices $i,j$ to designate a grid cell. An Ensemble Kalman Filter works by running different simulations, or ensemble members, simultaneously with variations in model parameters (e.g., initial conditions). We use $N_e$ members in the following.

#### 2.2.1 Projecting weights on densities

At each step t, we define:

– $w_t^f$ the forecast weights vector, of size $N_p$, in kg.

    – $x_t^f$ the forecast densities vector computed after projecting $w_t^f$ on the density grid, in $\mathrm{kg.m^{-2}}$.

    – $y_t$ the density observations vector, in $\mathrm{kg.m^{-2}}$, with its error covariance matrix $R$.

    – $x_t^a$ the analyzed densities vector computed by assimilating observations $y_t$ in $x_t^f$ via the Ensemble Kalman Filter, in $\mathrm{kg.m^{-2}}$.

– $w_t^a$ the analysis weights vector computed by projecting on $w_t^f$ the corrections computed on $x_t^a$, in $\mathrm{kg}$.

    – $\Delta_{i,j,t}$ the set of particles present at step t in grid cell $i,j$.

To start, for grid cell $i, j$ with area $A_{i,j}$, $x_t^f$ is computed with the formula:

$$(x_t^f)_{i,j} = \frac{\sum_{p \in \Delta_{i,j,t}} (w_t^f)_p}{A_{i,j}} \tag{1}$$

In the following, we omit sub-index $t$ when all the operations are performed at the same time step $t$.

### 2.2.2 Assimilating with the Ensemble Kalman Filter (EnKF)

Our assimilation step relies on the use of Ensemble Kalman Filtering, as described in Evensen (2003). This method is derived from Kalman Filtering and notably suitable to situations in which the model is not an easily invertible matrix (used in standard Kalman Filtering), and one cannot efficiently compute an adjoint (used in Extended Kalman Filtering).

Standard Kalman Filtering allows computing the analysis state using a single equation. In standard Kalman Filtering, the forecast state vector $x^f$ (in this case, the densities) and the analysis vector $x^a$ are linked with:

$$x^a = x^f + K(y - Hx^f) \tag{2}$$

$H$ is the observation matrix that maps the state $x^f$ to the observation space of $y$.

The Kalman gain matrix $K$ is defined by the following equation:

$$K = P^f H^T (H P^f H^T + R)^{-1} \tag{3}$$

$R$ is the observation error covariance matrix. $P^f$ is the forecast error covariance matrix. When using a Kalman Filter, $P^f$ is in principle meant to be computed from the previous state by application of the forward integration matrix operator, but this is generally too computationally expensive and impractical. Here, we use Ensemble Kalman Filtering, where the $P^f$ matrix computation is approximated by relying on an ensemble of simulations.

Ensemble members are different instances of our simulation with different initializations. For ensemble member $k \in [|1, N_e|]$, we write $x_k^f$ the forecast state vector, and $\overline{x}^f$ the ensemble average

$$\overline{x}^f = \frac{1}{N_e} \sum_{k=1}^{N_e} x_k^f \tag{4}$$

Accordingly, the computation of $P^f$ can be accomplished using the formula:

$$P^f = \frac{1}{N_e - 1} \sum_{k=1}^{N_e} (x_k^f - \overline{x}^f)(x_k^f - \overline{x}^f)^T \tag{5}$$

Each ensemble member $k$ is then updated using equation 2 with $x_k$ instead of $x$.

### 2.2.3 Projecting the density updates on particles

Several ways of projecting the density updates (step 3 in figure 1) can be thought of. In the Ocean Plastic Assimilator v0.2, we simply choose to update the weights by uniformly distributing the density correction ratio of a grid cell $i, j$ among the particles in the same box using this formula:

$$\forall p \in \Delta_{i,j}, (w^a)_p = \frac{(x^a)_{i,j}}{(x^f)_{i,j}}(w^f)_p \tag{6}$$

In this equation, $(x^f)_{i,j}$ cannot be null when a grid cell $i, j$ contains particles (see equation 1), except if all particles have null weights. While extremely unlikely (we did not encounter this phenomenon during our numerous tests), particles with exactly null weights have to be taken out of the simulation.

This heuristic was chosen primarily for its simplicity and its computational efficiency. The multiplicative approach also tends to prevent computing negative weights if the density analysis is lower than the density forecast.

Finally, for step 4 in figure 1, since the dispersion model changes particles positions but not their weights when integrating, the forecast weights at time $t + 1$ are:

$$w^f_{t+1} = w^a_t \tag{7}$$

### 2.2.4 Initialization

As stated by Evensen (2003) the Ensemble Kalman Filter requires the initial ensemble to sample the uncertainty in variables that we want to update with data assimilation. In this article, we focus on our method's ability to compute the correct total mass of particles drifting. For this reason, we normally distribute the members' initial total masses with a mean $\mu_e$ and standard deviation $\sigma_e$. If we write $M_k$ the initial total mass for ensemble member $k$, we thus have:

$$M_k \sim N(\mu_e, \sigma_e) \tag{8}$$

Finally, we attribute an initial weight of $M_k/N_p$ to each particle.

## 3 Implementation and test-case setup

This section presents the Python implementation of the aforementioned method, called Ocean Plastic Assimilator (v0.2). We then describe the Lagrangian dispersion model (OceanDrift) used to generate double gyre dispersion simulations and the experiments created with it to observe how our method performs in a controlled environment.

## 3.1 Python implementation of the Ocean Plastic Assimilator

This first implementation is coded in Python (see Peytavin (2021a) for the repository). It is meant as a standalone program, using as input a dispersal model output data, formatted as a netCDF4 dataset containing particle coordinates in a given space and time domain, along with their weights. It is assumed that the advection in the dispersion model does not depend on particle masses. In the more general case, one would have to run the model again after each assimilation time step, as a change of a particle mass could change its future trajectory.

Once loaded, the input weights are duplicated in $N_e$ arrays, and the program runs the assimilation scheme presented in the previous section in a time loop, taking observations from an input data frame at each time step. The Assimilator can also take one additional dispersion simulation output from which it samples observations to assimilate at each time step. This is the approach used in the following test-case.

This implementation leverages the use of arrays and the fact that we only use one simulation for all ensemble members to 130 perform vectorized computations for the computation of $P^f$, equations 1 and 6. It also allows computing $\Delta_{i,j,t}$ only once for all ensemble members. Some parts of the algorithm are also executed with the just-in-time compiler numba (see Lam et al. (2015)) in order to run faster.

This implementation allows our algorithm to perform each following test-case, repeated assimilation of 2 observation points during 2000 timesteps in a $(60, 40)$ gridded domain, in less than a half-hour on a modern laptop, using about 3GB of storage 135 and 2GB of RAM.

Running the Assimilator on a dispersion output and not inside a dispersion model allows it to work on outputs from different models, as long as the data is appropriately formatted. Future implementations could also offer the option of running online (i.e., embedded inside a dispersion model), which could allow more flexibility and possibilities, as discussed in section 6.2.3.

## 3.2 Double gyre plastic dispersion using the OceanDrift model

In order to create our test cases, we first need a dispersion model and a flow field. We chose the OceanDrift model from the Norwegian Lagrangian trajectory modeling framework OpenDrift (see Dagestad et al. (2018)). It was chosen mainly for its simplicity and the fact that OpenDrift embeds a module to generate a dispersion based on a 2D double gyre flow field.

This field consists of two gyres moving closer then farther away periodically in an enclosed area. It's a simple field but complex enough to stir and disseminate particles and is regularly used as a standard case to study time-varying flows, for 145 example in Guo et al. (2018). The evolving currents are generated using an analytical field[1]. The equations generating this 2D, time-varying, deterministic field are:

$$u = -\frac{d\phi}{dy} = -\pi A \sin(\pi f(x,t)) \cos(\pi y)$$
$$v = \frac{d\phi}{dx} = \pi A \cos(\pi f(x,t)) \sin(\pi y) \frac{df}{dx} \tag{9}$$

---

[1]https://shaddenlab.berkeley.edu/uploads/LCS-tutorial/examples.html#Sec7.1

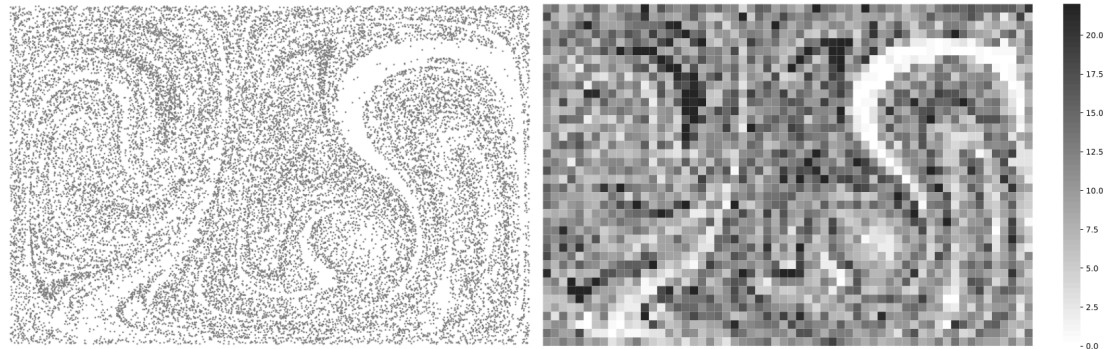

**Figure 2.** Generated particles (left panel) in a double gyre flow field with OpenDrift, and the corresponding plastics concentration field, in particles per grid area (right panel). The domain grid size is $60 \times 40$.

$$\begin{cases} f(x,t) = a(t)x^2 + b(t)x \\ a(t) = \epsilon \sin(\omega t) \\ b(t) = 1 - 2\epsilon \sin(\omega t) \end{cases} \tag{10}$$

The dimensionless domain size for these equations is $[0,2] \times [0,1]$.

Parameter $A$ is the circulation amplitude, $\omega$ is the frequency of oscillation of the gyres, and $\epsilon$ is the amplitude of the gyres oscillation relative to the steady-state.

Particles are then generated and advected using the OceanDrift lagrangian model from the Norwegian trajectory modeling framework OpenDrift (Dagestad et al. (2018)). Figure 2 shows such a dispersion and the associated concentration field.

Thus, we can generate different dispersion simulations by changing the initial particle positions seed, which changes the distribution of particles trajectories and the initial masses of the particles. We can also change the flow field parameters $A$, $\omega$ and $\epsilon$.

In the following section, we modify the flow field parameters and the particle positions seeds to create assimilation test cases that use two simulations: a reference and a forecast. We then sample observations from the reference simulation, and assimilate them inside the forecast simulation. By doing so, we mimic assimilating real concentration data into an uncertain flow field in the presence of model error.

### 3.3 Assimilation experiments setup

In order to assess and quantify the efficacy of the Assimilator in different cases, we designed two experiments.

The first one aims at verifying that, when the forecast flow field reproduces the reference flow field accurately, our implemented scheme can correct an incorrect total mass guess. It also intends to check that the estimate gets better when the observation error gets lower, as one would generally expect.

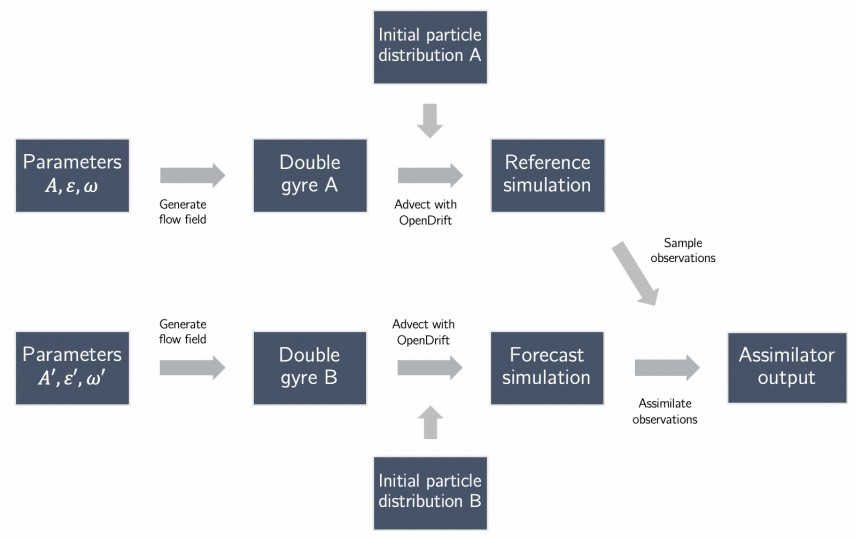

**Figure 3.** Schematic depiction of a test case using a reference and a forecast simulation

The second experiment aims to assess the Assimilator's behavior and efficacy when the forecast flow field is slightly different from the reference by changing the double gyre parameters $A$ and $\epsilon$.

In both experiments, we run several test cases to assimilate observations taken from a reference simulation into a forecast simulation using the Assimilator. Then, we compute the total plastics mass estimation error and the concentration field RMSE to assess how close the assimilated forecast gets to the reference situation. This procedure is depicted in Figure 3.

In each test case, the Ocean Plastic Assimilator is executed over the course of 2000 timesteps. The double gyre size, which is $[0,2] \times [0,1]$ is dimensionless, which means that the timestep is dimensionless too. However, if the flow field was the size of the great pacific garbage patch, then with $A = 0.1$ the timestep would be of the order of a day.

Over the double gyre, we define a gridded domain of size $(60, 40)$ and select two observation points, fixed, to run each assimilation test case. This sampling pattern can be thought of as representing a set of moorings that one may deploy in the real Ocean. $H$ is defined as a matrix that subsets $(x)_{i,j}$ to 2 points of observations.

For the $i$-th point, the measurement is simulated by adding a random error to $x_i$ such as :

$$y_i = \max(x_i + N(0, \sigma_{rel} x_i), 0) \tag{11}$$

To compute matrix $R$, we choose to model the observation error as a sum of an additive error $\sigma_0$ and a multiplicative, relative error $\sigma_{rel}$. As such, with $y_i$ the value measured at the $i$-th observation point:

$$R = \text{diag}(\sigma_0^2 + (\sigma_{rel} y_1)^2, \sigma_0^2 + (\sigma_{rel} y_2)^2) \tag{12}$$

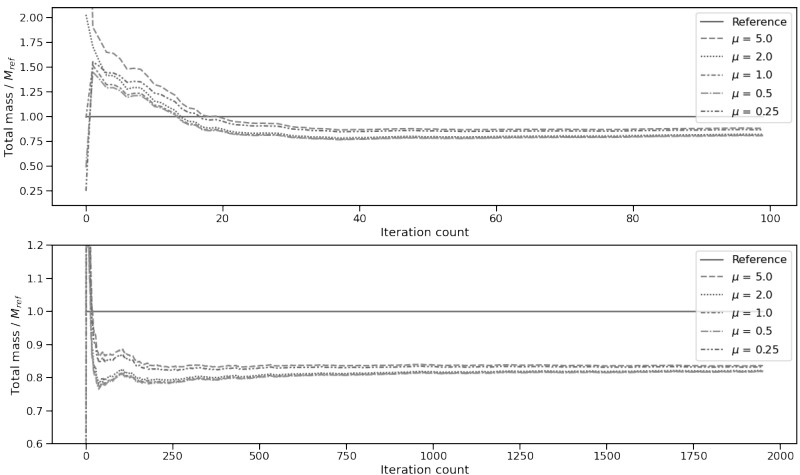

**Figure 4.** Evolution of total mass over time for five different forecast simulations with five different initial total masses (Tab. 1) over 100 assimilation iterations (top) and 2000 iterations (bottom). The total mass evolution of the reference simulation is indicated by a solid line.

In the following, unless specified otherwise, we use $N_e = 10$, $\sigma_e = 0.05$, $N_p = 25000$, $\sigma_0 = 0.1$ and $\sigma_{rel} = 1\%$. The 2 observation points coordinates are the following pairs: $(12, 4), (55, 27)$.

## 4 Results

### 4.1 Estimating the total plastics mass in the forecast

In this first experiment, we want to assess the ability of our newly implemented scheme to estimate the total mass of plastics in the reference simulation correctly.

First, we generate a reference situation using $\epsilon = 0.25$, $A = 0.1$ and $\omega = 2\pi/10$. We input the same parameters to integrate the particles trajectories in the forecast simulation. By doing so, we are in a position where we understand the flow of the reference situation correctly, but we do not know the total mass of plastics drifting. In the following, $M_{ref} = 25000$ is the constant, total mass of the reference situation.

We initiate 5 different forecasts with $\mu_e = 0.25 M_{ref}$, $0.5 M_{ref}$, $M_{ref}$, $2 M_{ref}$ and $5 M_{ref}$. Observations are collected (and later assimilated) at each time step on 2 observation points which could for example represent a pair of moored instruments.

Figure 4 shows the evolution of the forecast total mass for each simulation. Forecasts starting with an initial total mass lower than approximately $0.82 M_{ref}$ have their total mass rise while those starting with higher total mass have their total mass fall. Final total plastics mass in the forecast after 1900 steps of assimilation for each simulation are presented in table 1 . Overall,

| $\mu_e$ | FTM / $M_{ref}$ | RMSE$_f$ | RMSE$_\emptyset$ |
|---|---|---|---|
| 0.25 | 0.833 | 4.626 | 8.661 |
| 0.5 | 0.818 | 4.660 | 6.467 |
| 1 | 0.820 | 4.656 | 4.675 |
| 2 | 0.822 | 4.652 | 12.944 |
| 5 | 0.836 | 4.619 | 45.714 |

**Table 1.** Final Total Mass (FTM) relative to $M_{ref}$ and the concentration field RMSE for 5 different forecast simulations with 5 different initial total masses $\mu_e$. RMSE$_f$ and RMSE$_\emptyset$ are the concentration field RMSE at the end of simulations, with and without assimilation of observations.

| $\sigma_{rel}$ | FTM / $M_{ref}$ | RMSE$_f$ | RMSE$_\emptyset$ |
|---|---|---|---|
| 0.5 % | 0.895 | 4.546 | 12.944 |
| **1.0%** | **0.822** | **4.652** | **12.944** |
| 2.5 % | 0.728 | 4.981 | 12.944 |
| 10 % | 0.611 | 5.640 | 12.944 |

**Table 2.** Parameters and metrics for assimilation simulations with different values of $\sigma_{rel}$, with $\mu_e = 2$. $FTM$ is the Final Total Mass, RMSE$_f$ and RMSE$_\emptyset$ are the Concentration Field RMSE at the end of simulations, with and without assimilating.

the forecasts total masses seem to converge towards a similar value of approximately $0.82 M_{ref}$, from which we can conclude that in this situation, the method makes an $18\%$ error.

Another indicator of the correctness of a simulation can be computed from the concentration field at each step. For one of the forecasts (with $\mu_e = 2$), we analyze the distribution of concentration errors, over the gridded domain, and through time (figure 5). We observe a decrease in the mean absolute percentage error and a decrease in absolute percentage errors' standard deviation. We also observe that this distribution does not contain overly large values.

We also compute the concentration field Root Mean Square Error RMSE$_f$ at the end of the simulation after assimilating, and RMSE$_\emptyset$ at the end of a simulation with no assimilation. Values in Table 1 indicate a clear improvement of the RMSE when the initial total mass was erroneous and a stable one compared to no assimilation when the initial total mass was correct.

Overall, this points to an improvement in the forecast concentration field over time, thanks to data assimilation.

Finally, in order to assess the method accuracy depending on observation errors, we set $\mu_e = 2$ and run simulations with different values of $\sigma_{rel}$. FTM and RMSE are then computed and presented in Table 2.

We find that decreasing $\sigma_{rel}$ increases the final total mass of the forecast, getting it closer to 1 while the RMSE decreases. This demonstrates that the forecast bias can be reduced by decreasing the observation error, as one would usually expect of a data assimilation method.

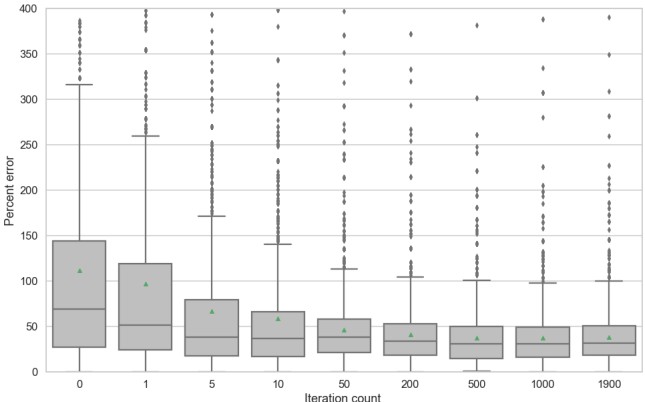

**Figure 5.** Evolution of the error field between the reference concentration field and the forecast concentration field, in percent, for $\mu_e = 2$. At each timestep, the error field is computed and the distribution of the absolute errors in each cell, in percent of the cell reference concentration, is depicted in the box plots. Dots outside whiskers represent outliers and the triangle is the mean.

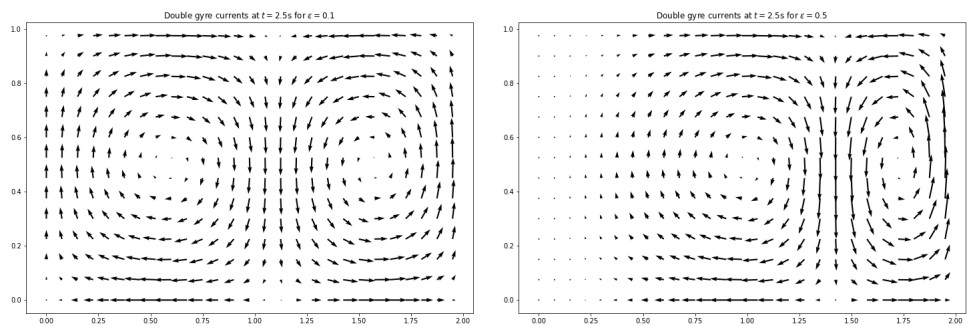

**Figure 6.** Flow fields at $t = 2.5$s for two double-gyre simulations with (a) $\epsilon = 0.1$ and (b) $\epsilon = 0.5$

## 4.2 Impact of physical model errors

In this second experiment, we change the parameters used to generate the currents of the reference simulation double gyre. For example, the impact of a modification of $\epsilon$ on the generated flow field is illustrated in Figure 6. By assimilating observations from reference situations with different double-gyre parameters, we can observe the effects of having an erroneous physical dispersion model when assimilating data.

We initiate the forecast with an erroneous initial total mass of $2M_{ref}$ and expect that the best total mass predictions will arise from assimilation simulations with the closest flow field.

The forecast simulation is generated using $\epsilon_{ref} = 0.25$, $A_{ref} = 0.1$ and $\omega_{ref} = 2\pi/10$.

| $A$ | $\epsilon$ | FTM / $M_{ref}$ | $\text{RMSE}_f$ | $\text{RMSE}_\emptyset$ |
|---|---|---|---|---|
| **0.1** | **0.25** | **0.822** | **4.652** | **12.944** |
| 0.105 | 0.25 | 0.810 | 4.871 | 13.037 |
| 0.11 | 0.25 | 0.752 | 5.249 | 13.204 |
| 0.125 | 0.25 | 0.744 | 5.658 | 13.455 |
| 0.1175 | 0.25 | 0.733 | 5.718 | 13.444 |
| **0.1** | **0.25** | **0.822** | **4.652** | **12.944** |
| 0.1 | 0.3 | 0.781 | 5.507 | 13.293 |
| 0.1 | 0.5 | 0.770 | 5.170 | 13.402 |
| 0.1 | 1.0 | 0.738 | 5.897 | 13.789 |
| 0.1 | 0.0 | 0.276 | 29.241 | 30.856 |

**Table 3.** Parameters and metrics for simulations with different values of $A$ and $\epsilon$ for the reference simulation. $FTM$ is the Final Total Mass, $\text{RMSE}_f$ and $\text{RMSE}_\emptyset$ are the Concentration Field RMSE at the end of simulations, with and without assimilating.

We then generate different reference simulations with different values of $A$ and $\epsilon$, and try assimilating observations sampled from each of them into the forecast.

We find that data assimilation remains effective and that simulations run with values of $\epsilon$ and $A$ closer to $\epsilon_{ref}$ and $A_{ref}$ lead to better estimations of the total mass and concentration field after some time as one might expect (Figure 7 and Table 3).

This result illustrates that the assimilation method can be robust to unknown model errors.

## 5 Application to the Great Pacific Garbage Patch

In this section, we present an application to real-world global dispersion models. As before, we sample observations from one simulation and assimilate them into another in order to mimic the assimilation of observations that could be collected daily by a pair of moorings deployed in the real Ocean. We just use an estimate of real ocean currents in place of the simplified double

gyre defined in equation 9.

We generate two global dispersion simulations with the Lagrangian Dispersion Model presented by Lebreton et al. (2012). In both cases, the circulation model uses output from the HYbrid Coordinate Ocean Model (see Bleck (2002)), available every 6 hours at 0.08 degrees. This estimate includes Ekman transport and their convergence, as well as mesoscale eddies. The first simulation has particles seeded along the coasts of 192 countries depending on reported garbage input estimates. The second

simulation has particles seeded at rivers mouths only, based on estimates of their plastics outflow. Both generation models are described in the supplementary materials of Lebreton et al. (2018). A model spin-up was done from 1993 to the end of 2011.

We initialize plastic particles masses generated in the coastal seeded model depending on their release year. If $x$ the time spent (in fraction of years) since the beginning of the simulation, then $w_p = 1 + x + \frac{1}{2\pi}\sin(\pi(2x+1))$ is the mass of particles, in tonnes, seeded at time $x$. This formula increments particle masses by 1 tonne each new release year, with some periodic

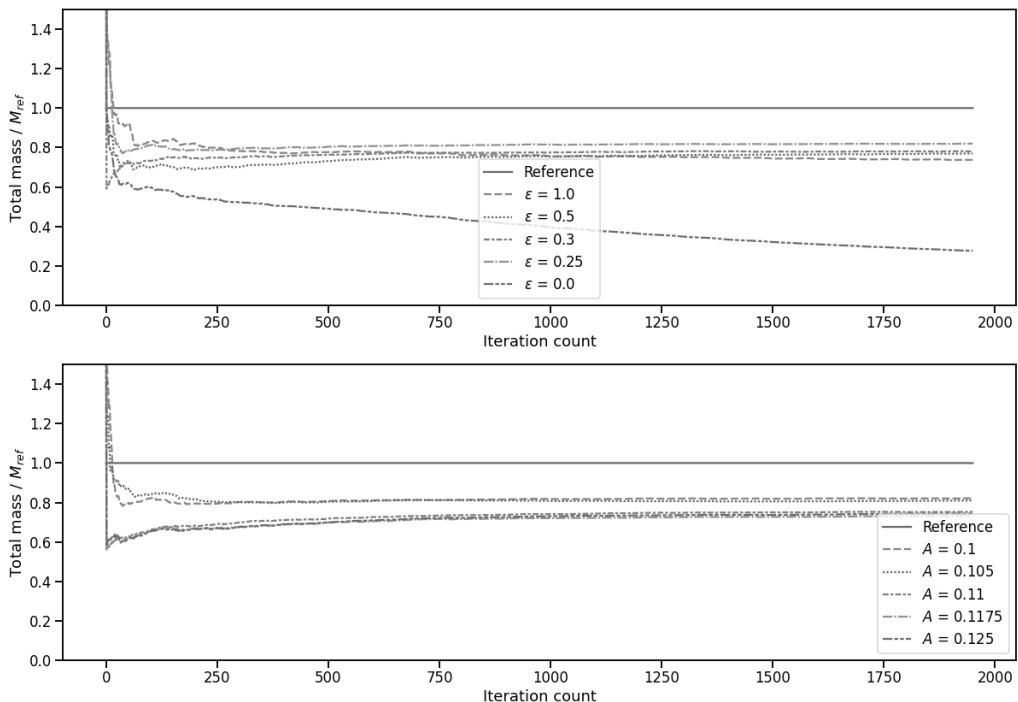

**Figure 7.** Evolution of the total plastics mass in the forecast simulation for 5 different runs with varying values of double gyre parameters $A$ and $\epsilon$, along with the total plastics mass in the reference simulation.

variability. The particles masses in the rivers-seeded simulation are initialized to 1 tonne regardless of their release date. By doing so, we mimic a situation where we underestimate the yearly increase of plastics mass input into the ocean.

The gridded domain has a resolution of 0.5 degrees, with 80 by 44 points, going from 165° W to 125° W and from 23° N to 45° N. Throughout 2012, we sample two observations per day at positions 152.5° W, 29° N and 140° W, 35° N from the coastal-seeded dispersion simulation and assimilate them in the rivers-seeded dispersion model. We use $N_e = 10$, $\sigma_e = 50$, $N_p = 25000$, $\sigma_0 = 0.1$ and $\sigma_{rel} = 1\%$.

Our method is able to predict the total mass of floating plastics with a 17% error, and to divide by 4 the concentration field RMSE (figure 8). The computations take about an hour to run on a standard laptop.

The updates to the concentration field are presented in figure 9, which shows that, as expected, the assimilated forecast has increased concentrations.

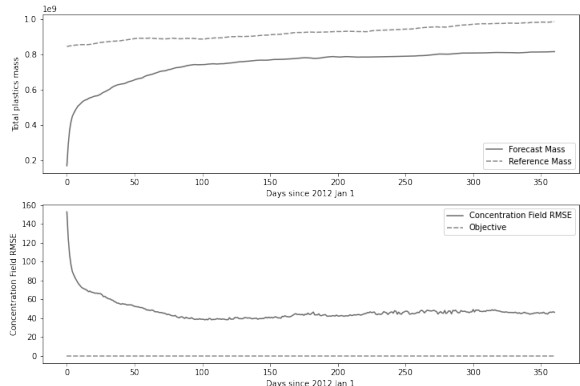

**Figure 8.** Top : Evolution of total plastics mass in the domain through 2012, for the reference simulation and the forecast simulation. Bottom : Evolution of the concentration field RMSE in the assimilation domain through year 2012.

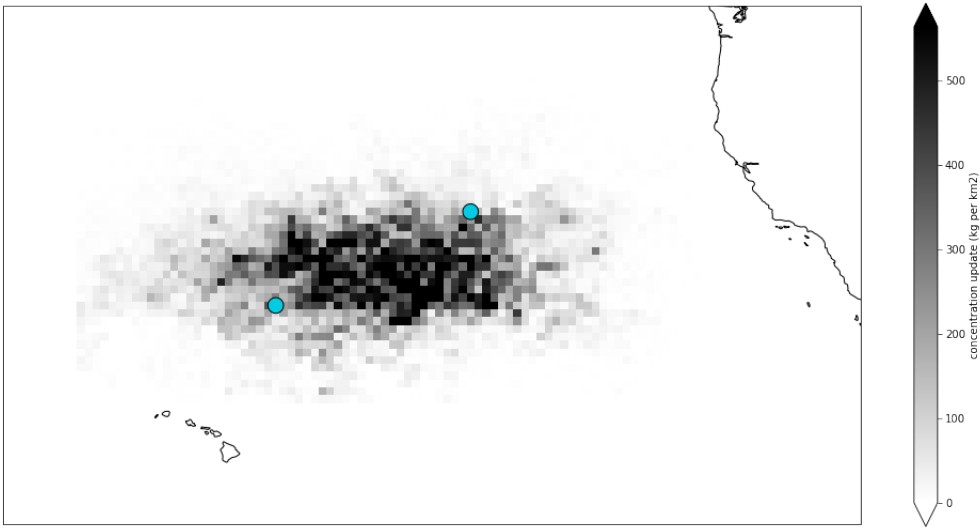

**Figure 9.** Concentration field updates at the end of the assimilation cycle, with the two observation locations in blue. This field is the difference between the forecast concentration field at the end of the year 2012 with assimilation, and the same without assimilation.

Further experimentation will be required to assess the benefits of using this method in real-world use cases, with real data. However, these results confirm the potential skill of our method, even in the presence of sizable model error.

## 6    Discussions and perspectives

### 6.1    Towards an application to real-world data

In this proof-of-concept paper, we placed ourselves in a controlled environment to assess the efficacy of the method. In the
future, our goal will be to eventually apply the method to real data by replacing the simulated reference situation observations with real-world observations, and the previous results can help in understanding what might happen in assimilating real-world data. The fact that replacing the analytic circulation field by a real-world one (in section 5) did not prevent the method from improving the forecast is viewed as an encouraging first step in that direction.

In figures 7 (a) and (b) we observed that the more accurate the underlying dispersion model is, the more accurate the
assimilation result is. For our method to be applied successfully to a real global plastics assessment model, its dispersion prediction would have to be accurate enough. Ongoing work which is focused on identifying model error sources and refining statistical priors should benefit the planned application to real data (e.g. Maximenko et al. (2012), van Sebille et al. (2020), Meijer et al. (2021)).

Conveniently, we observed that the forecast total mass gets higher when the dispersion model is more accurate, thus acting,
in a way, like a score. As a result, we might discriminate between dispersion models based on this method's output by selecting the ones that output the highest total mass.

### 6.2    Future Developments

Amongst the potential applications of the presented method, one might highlight the evaluation and design of real observational strategies. Here we considered one hypothetical, albeit plausible, scenario which might represent the deployment of a few
relatively accurate moorings. In future studies it would be interesting to investigate how data coverage in space and time may affect forecast skill in more detail, for example, or use this data assimilation system as a benchmark for proposed field campaigns. Several directions to further develop the method and make it more accurate also seem worth considering, as outlined below.

#### 6.2.1    Improving the filter

Throughout the last two decades, the Ensemble Kalman Filter has been extensively developed and improved, with numerous variants published in the scientific literature. Using different ensemble sampling strategies or a square-root algorithm was described as a way to improve accuracy in Evensen (2004). Other solutions include inflating the ensemble before assimilating (see Anderson (2007)), resampling the ensemble, or using a method to assimilate observations locally by adding a Schur product with a so-called correlation matrix in the computation of the Kalman gain in equation 3 (see Houtekamer and Mitchell

(2002)). Assimilating locally around observation locations could also have the advantage of further improving the geography of the concentration field, which would translate in reduced values of $\text{RMSE}_f$.

### 6.2.2 Decoupling the ensemble members particles positions

The method presented here uses the same dispersion simulation as a base for all the ensemble members particles trajectories. In all members, the particles positions through time are the same, the only variables that differ are the particles masses. In
particular, the particles trajectories are the same in each member. This approach reduces greatly the storage cost and increases computation speed.

However, it significantly lowers the diversity of the ensemble, so in future work one might want to decouple the ensemble members trajectories, i.e have for each member a unique set of trajectories. We anticipate that extending the method to use an ensemble with diverse particle simulations should help the forecast converge towards a concentration field closer to the
290 reference one. We regard this possibility as a leading candidate to make the method even more accurate.

### 6.2.3 Studying other projection operators

In section 2.2.3, we presented a simple way to update particles weights after assimilating density observations through the equation 6. Different possibilities of performing this step have been thought of, some of which we think may be worth investigating further. Another simple approach would be to apply an additive correction, instead of the multiplicative correction used
in equation 6:

$$\forall p \in \Delta_{i,j}, (w^a)_p = (w^f)_p + \frac{((x^a)_{i,j} - (x^f)_{i,j})}{card(\Delta_{i,j})} \tag{13}$$

This approach was not favored in this first study, as it seemed more likely to generate negative weights more often.

Another alternative would be to generate new particles so that their weights sum up to the updated density, possibly fewer or more particles. This could be more technically challenging to implement and require implementing the assimilation scheme
directly inside the dispersion model loop. However, it could also have the advantage of conveniently increasing resolution where there are high plastics concentrations.

## 7 Conclusions

This paper presents a simple yet readily effective method to assimilate observations of plastics concentration data into a Lagrangian dispersion model, and its first implementation called the Ocean Plastic Assimilator (v0.2). We apply it in a controlled
environment to assess its efficacy. We study the impact of observation errors on the prediction accuracy and changed some of the dispersion parameters ($A$ and $\epsilon$) to evaluate the impacts of model errors. Finally, we apply the method to a more realistic case with a real-world circulation field and find that the method still performs well. The encouraging results indicate that it is an excellent candidate to perform data assimilation with real-world data over ocean gyres.

Thus, the Ocean Plastic Assimilator will be further developed at The Ocean Cleanup to assimilate plastics concentration data from the oceans and improve our cleanup operations in oceanic gyres. This method will undergo more research to develop its features and assess its efficacy when using real-world observations. We expect it to be used to assess in real-time the cleanup operations of The Ocean Cleanup.

The simplicity of the developed data assimilation method means that it should be easy to generalize to various other popular open-source lagrangian frameworks such as OceanParcels (Delandmeter and van Sebille (2019)) or MITgcm (Campin et al. (2020)). Porting the data assimilation procedure to the Julia language is also being envisioned whereby one could leverage the newly developed IndividualDisplacements.jl package to carry out Lagrangian simulation of plastics concentrations (Forget (2021)).

*Code and data availability.* The version of the model used to produce the results presented in this paper is archived on Zenodo (Peytavin (2021a)), as are the input data to run the model and the raw data presented in this paper (Peytavin (2021b)). The code repository contains a Python notebook that allows to download necessary data and reproduce the presented experiments.

*Author contributions.* B-S.R. conceived and presented A.P. the idea of applying Data Assimilation to a dispersal model. A.P. studied the Data Assimilation literature and suggested using an Ensemble Kalman Filter. A.P. wrote and maintained the code, and applied it initially to real oceanic data. J-M.C. introduced A.P. to G.F. and G.F. with B-S.R. recommended applying the method on an analytical flow field to assess its performance. A.P. structured the manuscript, wrote the initial draft and the next versions, and prepared figures. G.F. and A.P. met every two weeks or so to discuss the manuscript as A.P. was writing it, G.F. provided numerous advice on tweaking the method and improving the manuscript. B-S.R. and J-M.C. were sometimes also present to provide advice during these meetings.

*Competing interests.* The authors declare that no competing interests are present.

*Acknowledgements.* Axel Peytavin and Bruno Sainte-Rose would like to thank The Ocean Cleanup and all its funders for supporting them. Gael Forget acknowledges support from NASA-IDS award 80NSSC20K0796 , NASA-PO award 80NSSC17K0561 , and the Simons Foundation award 549931. The authors acknowledge the reviewers for their careful reading of our manuscript and their comments.

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
