# Peer review of "Ocean Plastic Assimilator v0.2: Assimilation of Plastics Concentration Data Into Lagrangian Dispersion Models."

_Geoscientific Model Development, 2020_

## Author Comment (AC1)

We thank the reviewers for their helpful comments and suggestions, which helped improve our manuscript. In response to reviewer 1, we added a small section with a more realistic ocean circulation test case which uses a simulation of the North Pacific. We also streamlined a few things as suggested by Reviewer 2 in particular.

Please find detailed responses, point by point, below.

**Responses to reviewer 1**

1) *The implementation of EnKF focuses on estimating the weights of individual particles, yet as far as I understood the trajectories of these particles are not data-assimilated. I would argue that most uncertainty in Lagrangian simulations of floating plastic is on the processes affecting their pathways (windage, mixing, etc), so would a EnKF implementation that also adapts the parameters of the particle properties beyond weight not be much more relevant? The authors do mention this briefly in the outlook, but this could certainly be discussed further (especially to what extent the method has to be adapted then).*

We agree with the reviewer that alternative methods are worth exploring and this might help improve upon what's achieved here. We decided to focus on particle weights first given that this readily provides a suitable way to assimilate concentration measurements. Indeed, most of the measurements located in space and time that we have are concentration measurements, which can easily be assimilated in particle weights. We suspect that it would be more complicated to assimilate concentration measurements via other particle properties, either in a Eulerian or in a Lagrangian frame, but various ideas could indeed be worth exploring beyond the scope of this article.

The text on this was modified for clarity and we added references to recent assessments of the various sources of model error in advection processes and address this reviewer comment. Please see also our response to the next comment.

2) *The project could fairly easily be advanced a level beyond this proof-of-concept by also including a slightly more realistic test case on for example a simulation of the North Pacific. Does the code still perform well then? In particular, the double gyre is an incompressible flow so the particle density is (roughly) uniform in space (see also Fig 2). This is very different from the North pacific gyre, where plastic concentrations vary by orders of magnitude because of Ekman convergence. How good is the method in such a more complicated system?*

We would like to thank the reviewer for this suggestion. We performed one such test on a simulation of the North Pacific generated with currents of HYCOM (which includes Ekman convergence and eddies). We found similar performance (about 17% of error on the forecast of the total mass at the end of the simulation). The computational time did increase, up to an hour to assimilate daily for 365 days, which is still reasonable for real use cases. This test case and the results are presented in a new

section (section 5), between lines 226 and 251. We consider that a more detailed and thorough investigation would be important and useful but can be left for further study.

3) *I wonder how sensitive the results are to the choice of the two 'observation' points. Why have they been chosen where they are? Are these particularly 'good' or 'bad' locations for the EnKF? This should be discussed*

We observed that the results do not vary much when we change the observation points. The two points chosen for the paper were selected a priori — ie., not because they were specifically good or bad. To double check, we ran several new simulations with all identical parameters except the assimilation location and found similar results, see below. We considered adding these new results as an appendix to the paper (see image below) but thought that they could also be left out of the scope of this paper. The optimal design of an observation array is an important goal but not one we are aiming to achieve in this paper.

| Observation locations | FTM / $M_{ref}$ |
| --- | --- |
| $(15,5), (55,35)$ | 0.833 |
| $(\mathbf{12,4}), (\mathbf{55,27})$ | **0.822** |
| $(40,12), (24,28)$ | 0.817 |
| $(30,20), (25,25), (40,15)$ | 0.800 |
| $(35,15), (23,37), (53,22), (14,30)$ | 0.798 |
| $(35,15), (23,37), (53,22)$ | 0.775 |
| $(25,10), (30,10)$ | 0.766 |
| $(12,12), (10,30), (30,20), (45,15), (5,5)$ | 0.759 |

**Table 1.** Final Total Mass (FTM) relative to $M_{ref}$ for 8 different forecast simulations with 8 different sets of assimilation locations.

**Impact of the choice of observation points**

In section 4, we present results of our double gyre test case using assimilation locations at coordinates $(12,4), (55,27)$. In order to assess whether our findings were dues to a particular choice of observation points or not, we ran other simulations while only changing the points coordinates and their number. We again use $N_e = 10$, $\sigma_e = 0.05$, $N_p = 25000$, $\sigma_0 = 0.1$, $\sigma_{rel} = 1\%$ and $\mu_e = 2M_{ref}$. The final total plastic mass are computed and presented in table 1.

While these results show that the forecast of the total mass prediction depends on the choice of the observation locations, no clear correlation between the number of observation points and the forecast could be determined at first glance. Optimizing the selection of observation locations can thus be considered an open problem. However, we can conclude that the observation points randomly selected for our test cases did not produce statistically unusual results. As such, the results of section 4 represent what an user of the Ocean Plastic Assimilator v0.1 could expect to get.

4) *The implementation of EnKF by the authors requires constant regridding of the particle locations to a Field. This is an expensive step, that I'm not entirely sure is needed. Would it not be possible to do the EnKF directly on the particle positions themselves and work entirely in the Lagrangian framework?*

We agree that exploring the direct assimilation of positions could be a useful approach. However, the method we chose is simple and seems rather efficient. Thus, we needed minimum development time to reach a point where the data assimilation runs on a standard laptop and produces useful results. We actually find that to be remarkable. Therefore, we decided to focus on analyzing results for this paper rather than do further technical developments at this time.

**Responses to Reviewer 2**

*General comments: The paper describes an interesting topic with a relevance for present day clean-up of plastics waste distributed in the oceans and is well written. This study is only a proof-of-concept and the main focus of the work is the assimilation of observations into a LG dispersion model to improve the forecast of plastics distribution in the ocean. Although the authors study the sensitivity of the total plastics mass to certain parameters of the flow field and the values of the initial plastics mass in detail, the sensitivity of the results to location and the number of observational points is, however, not considered.*

*This paper is recommended for publication after some corrections.*

We thank the reviewer for their useful comments and their overall support of this paper.

With regard to the sensitivity of the results to the choice of the observation points, please refer to the response to question 3 of reviewer 1, in which we show that we did not choose particularly "good" or "bad" observation points, but ones that are representative of what we could expect with this method. A more detailed study of the design of observing systems is beyond the scope of the paper however, in our view.

In the following, we answer each specific comment

*Page 1, line 7: What is meant by "more suitable"? Can you describe what you mean more specifically?*

Was indeed not clear and has been rephrased by "closer to the real circulation".

*Page 1, line 8: "can effectively be approached": do you mean: can effectively be improved by this simple assimilation scheme?*

Yes, we rephrased as suggested.

*Page 1, line 13 and 19: Please cite the publication of Lebreton et al. correctly, see also the references chapter:*
*Lebreton, L., Slat, B., Ferrari, F. et al.: Evidence that the Great Pacific Garbage Patch is rapidly accumulating plastic. Sci Rep 8, 4666 (2018). https://doi.org/10.1038/s41598-018-22939-w*

Done

*Page 2, line 37: Please clarify which of the following sub-sections belong to the Plastic Assimilator v0.1 method by summarizing it under the respective section with that name.*

All the processes presented in section 2 are included in the Ocean Plastic Assimilator v0.1, except the alternative methods being envisioned in section 2.2.3. We clarified this point in section 2.2.3.

*Page 2, line 38: Instead of "This" I would suggest "It".*

Done

*Page 2, line 39 (see also line 54): Replace "how we go back and forth" by "... transformation between Eulerian and Lagrangian space ..."*

Done

*Page 2, line 40: I would delete the last sentence of this paragraph.*

Done

*Page 3, line 57: ..gridded domain, with a grid size (m,n), and indices i,j to ...*

Done

*Page 4, line 73: I would suggest to write out "ensemble Kalman filter (EnKF)" in the section heading.*

Done

*Page 5, line 112: "then"? This sounds as if the initial weight is a consequence of eq 8! Please comment.*

Indeed, it was choice we made and not a consequence of eq 8. We rephrased.

*Page 6, line 120: "Advection scheme of the dispersion model"? Do you mean "advection in the dispersion model"?*

Yes, we replaced by the suggestion

*Page 6, line 121: what do you mean by "regenerate the particle trajectories"?*

Was rephrased.

*Page 7, line 156: "use these possibilities"? Do you mean "modify the flow field parameters and the particle positions seeds .... to create...*

Yes, we rephrased.

*Page 7, line 157: Please rephrase the sentence: "By using plastics concentrations sampled from the reference simulation and assimilated into the forecast simulation, we can mimic assimilating observational plastics data ... errors."*

Done

*Page 8, line 167: Please be more specific and describe the "metrics".*

Done, we meant total plastics mass and RMSE.

*Page 8, line 172: How sensitive are your results to the choice of the observation points with respect to number and location in your analysis?*

Please see responses already provided above.

*Page 9, Figure caption 4: "assimilation simulations" = "forecast simulation"? I would suggest to modify the text of the figure caption to: "Evolution of mass over time for five different forecast simulations with five different values of initial total mass ... The mass evolution of the reference simulation is indicated by a solid line."*

Done

*Results section: In several places (Fig.4, Sec. heading 4.1, text) you use the term "forecast/reference mass". This sounds odd to me. What you actually mean is the evolution of plastics mass in the forecast/reference simulation. Would you please change the wording where appropriate? Is there a difference in meaning when you use "total mass" instead of "mass"?*

There is indeed no difference in meaning. For clarity we rephrased everything to "total mass".

*Page 10, Table 1: "Final total mass (FTM) relative to M_ref and the concentration field RMSE for 5 different forecast simulations ... with and without assimilation of observations."*

Done

*Page 10, line 197ff: How do you calculate the concentration errors? You describe a decrease in the mean absolute error and reference to figure 5. Figure 5, however, displays a percentage error (what does this mean?). Please rephrase this paragraph and the description of the figure 5 caption.*

We added complementary information in the caption

*Page 14, line 250ff: Could you please explain how the ensemble member particles are coupled when they use the same dispersion simulation? Please, elaborate on this.*

We hope to have clarified this in the paper with the following (lines 284 to 288).

In all members, the particles positions through time are the same, the only variables that differ are the particles masses. In particular, the particles trajectories are the same in each member. This approach reduces greatly the storage cost and increases computation speed. However, it significantly lowers the diversity of the ensemble. We could decouple the ensemble members trajectories, i.e have for each member a unique set of trajectories.

***Page 14, line 265: "..method to assimilate observations of plastics concentration data into.."***

Done

***Page 14, line 271: Thus, it ....? the Ocean Plastic Assimilator?***

Yes. Rephrased.

***Page 14, line273: "on real-world"... do you mean: using real-world observations. ?***

Yes. We rephrased as suggested.